# Survey of the effect of viewing an online e-cigarette advertisement on attitudes towards cigarette and e-cigarette use in adults located in the UK and USA: a cross-sectional study

Paula Booth,[ ][1] Ian P Albery,[2] Sharon Cox,[2] Daniel Frings[2]

[1]Psychological Sciences Department, Applied Health and Communities College, School of Psychology, The University of East London, London, UK
[2]Division of Psychology, School of Applied Sciences, London South Bank University, London, UK

**Correspondence to**
Dr Paula Booth;
p.booth@uel.ac.uk

## ABSTRACT

**Objectives** This study explored the potential for e-cigarette advertisements to (1) enhance attitudes towards cigarettes and/or (2) reduce barriers to e-cigarettes uptake. The study tested whether exposure to an online electronic cigarette advertisement changed attitudes towards cigarettes and e-cigarettes in smokers, non-smokers, e-cigarette users and dual users (smokers who also use e-cigarettes).

**Design** Cross-sectional study

**Setting** Online survey

**Participants** Adults (n=964) aged 18 to 65 years old (M=36 years, SD=11.6) from the UK and USA. Participants were grouped into current non-smokers, e-cigarette users, dual users and smokers.

**Interventions** Participants viewed 1 of 15 randomly assigned online e-cigarette advertisements.

**Primary measures** Three single seven-point Likert scales measuring health, desirability, social acceptability were completed pre and post advertisement exposure.

**Results** Post exposure all smoking groups showed a decrease or no change in how socially acceptable or desirable they rated cigarettes. Paradoxically, dual users rated cigarettes as being significantly healthier after viewing the advertisement (p=0.01) while all other smoking group ratings remained the same. There was an increase or no change in how all smoking groups perceived the healthiness and desirability of e-cigarettes

**Conclusions** We observed no evidence that exposure to an e-cigarette advertisement renormalises or encourages smoking in smokers, non-smokers or e-cigarette users. However, there is some indication that viewing an e-cigarette advertisement may increase duals users' perceptions of the health of smoking.

## INTRODUCTION

There is a large body of evidence to support the notion that advertising cigarettes encourages people to start or to continue to smoke.[1][2] With the intent of reducing tobacco use globally, Article 13 of the WHO Framework Convention for Tobacco Control calls for 'a comprehensive ban on advertising,

---

| Strengths and limitations of this study |
| --- |
| ► This study is among the first to examine the attitudinal processes underlying the effect of viewing an advertisement on interest in both smoking and vaping. |
| ► The sample was derived of smokers, non-smokers, vapers and dual users rather than focusing on only one smoking type which is often a limitation of the extant literature in this area of research. |
| ► The effects of viewing an advertisement on actual smoking behaviour was not assessed and should be considered in future work. |
| ► Multiple analyses increased the risk of type 1 errors. |
| ► The effects of different modalities of advertising were not assessed. |

promotion and sponsorship of cigarettes'[3] (p. 11). However, this treaty does not cover e-cigarettes. While there are restrictions on advertising e-cigarettes in the UK (Committee on Advertising Practice)[4] and in the USA (US Food and Drug Regulations)[5] e-cigarettes are still being marketed elsewhere and there is concern that the advertisements portray e-cigarettes as being glamourous.[6] In these territories, the public now experiences increased exposure to e-cigarette advertisements[4][7][8] in the form of television, magazines, newspapers, billboards and internet advertisements.[9] The current paper explores the potential of such advertisements to (1) enhance attitudes towards cigarettes and (2) reduce barriers to e-cigarette uptake among current smokers.

One concern related to e-cigarette advertisements is that they may renormalise smoking. The denormalisation of smoking has been a cornerstone of tobacco control policies in encouraging existing smokers to quit and also to discourage initial uptake[10–12] and social acceptance of smoking has been

shown to be a barrier to reduction in smoking prevalence in a variety of communities.[13 14] It is therefore important to determine whether e-cigarette advertising may also affect attitudes towards cigarettes by making them more positive. A number of researchers have suggested that e-cigarette advertising could lead to increases in how socially acceptable and desirable cigarettes are perceived to be, and subsequently influence their continued use or possible (re)uptake in smokers, e-cigarette users and dual users (those who use both cigarettes and e-cigarettes).[6 15–17] Alongside observed effects on these groups, concerns that e-cigarette use may be a gateway to smoking suggest it is also important to determine whether advertising may encourage non-smokers to use e-cigarettes.[16 17] In this context, it is important to note that the evidence base around actual usage statistics currently suggests that e-cigarette availability does not appear to have led to regular use of either e-cigarettes or smoking among British non-smokers: the current number of UK non-smokers using an e-cigarette daily is less than 1%, and smoking prevalence rates continue to decline.[18] A similar pattern is observed in the USA.[19] However, results from laboratory studies exploring the effects of advertisements on urges to smoke and interest in smoking among other groups are inconsistent, and none have examined attitudinal processes which may underlie those effects which have been observed.[20–22] The current study aims to examine the issue of renormalisation by measuring attitude change towards cigarettes in adult non-smokers, smokers, e-cigarette users and dual users before and after viewing an advertisement.

In contrast to concerns about renormalisation, e-cigarette advertisements may represent an opportunity to promote reduced risk products as an alternative to smoking. Although there are currently only relatively few studies comparing the health effects of e-cigarettes use to smoking or non-smoking over time,[23 24] in the UK, e-cigarettes are widely supported as a reduced risk nicotine alternative for smokers, and has received support from a number of agencies.[18 25–27] However, in the UK, despite increasing public agency support, one barrier to the uptake of e-cigarettes among smokers has been an increasingly negative attitudes towards the products. Between 2015 and 2017, the proportion of UK smokers who believed that e-cigarettes are less harmful than cigarettes reduced from 31% to 20% (Action on Smoking and Health).[27] Similarly, in the USA, the percentage of smokers who perceived e-cigarettes to be equally harmful than combustible cigarettes increased from 23% in 2012 to 35% in 2015.[28] To the extent attitudes predict behaviour, such negative perceptions of the health of e-cigarettes are likely to reduce the potential for e-cigarettes to be used as a smoking cessation aid. E-cigarette advertising may serve to halt or reverse this decline. Research examining the effects of advertising on perception of e-cigarettes (carried out predominantly on younger populations) shows that viewing an advertisement may increase the perception of social acceptability and the intention to try an e-cigarette

in the future.[7 29] For instance, 13–17-year-old smokers showed a more positive attitude towards e-cigarettes after viewing an e-cigarette advertisement,[29] and adult smokers reported a decrease in the acute urge to smoke[30] and an interest in trying e-cigarettes.[22 31 32] Thus, in the current study, we also explore the potential for e-cigarette advertisements to change the way smokers (and other groups) perceive the healthiness, socially acceptability and general desirability of e-cigarettes.[13 33]

One limitation of the extant literature in these areas is that it focuses on either smokers, non-smokers and/or e-cigarette users. This leaves one important population, dual users, understudied. In a survey of e-cigarette use in Europe, of an estimated 37 million e-cigarette users, 73% were currently still smoking.[34] Many dual-users use e-cigarettes in areas in which they are unable to smoke cigarettes.[35] However, they still have a preference for smoking cigarettes in stressful situations or for pleasure.[36] Thus, it is important to establish whether viewing an e-cigarette advertisement can change attitudes towards smoking in dual-users.

To explore the potential for-cigarette advertisements to (1) enhance attitudes towards cigarettes and/or (2) reduce barriers to e-cigarettes uptake, the current study investigated whether viewing an e-cigarette advertisement influenced the extent to which smoking and e-cigarettes were perceived to be more or less healthy, desirable and socially acceptable in dual users, smokers, non-smokers and e-cigarette users.

## METHODS
### Design
This study had a cross-sectional design in which attitude measures were taken pre and post advertisement viewing. The between-subjects factor was smoking group (non-smokers, e-cigarette users, dual users and smokers). The dependent variables were the scores on a series of questions rating attitudes around the healthiness, desirability and social acceptability of both e-cigarettes and cigarettes.

### Participants, recruitment and procedure
Originally, 964 participants completed the survey, between December 2015 and February 2016, but data from 199 were removed for being under the age of 18 or not entering their age, failing controls (see below) and having missing data. Data were included for 765 participants, 361 men, 400 women and four others from an age range of 18 to 65 years old (M=36 years, SD=11.6). Participants were defined as smokers (n=115), non-smokers (410), e-cigarette users (100) or dual users (145). They were located in the USA (n=543) or the UK (n=222).

Participants were recruited through an online crowdsourcing tool (Crowdflower, similar to MTurk). Crowdflower users were given information about the survey on the Crowdflower site before being given the opportunity to proceed by giving online consent to participate in the

study and then clicking on the link to be transferred to a survey (delivered using Qualtrics). Crowdflower has an estimated pool of over 10 000 workers worldwide but response rates were unable to be calculated due to the age (18 to 65 years old) and location (UK or USA only) restrictions which would reduce the number of possible participants available.[37] Participants were paid a fee of $1.00 to complete the survey. Control questions were built into the Qualtrics survey to exclude automated responses and participants who were not concentrating appropriately on the questions (eg, 'For this item please indicate strongly agree'). Additional controls were set up on the survey so that participants were unable to enter the site from the same IP address more than once and that they had to take a minimum time of 90 s to complete the survey. Post hoc analysis indicated that a minimum of 100 participants (the smallest smoking group) was sufficient to detect an effect size r=0.15 or greater, (with Z=2.2 or above).

## Patient and public involvement

Patients were not involved in the development or conduct of this study. There are plans to disseminate the results to practitioners by including the results as an aspect in London South Bank courses provision aimed at service delivery managers and counsellors. A summary of the findings will be offered to Cancer Research UK and other policy organisations and promoted on relevant staff social media sites.

## Measures

### Attitudes towards cigarettes

Participants rated on three single measures how *healthy*, how *desirable* and how *socially acceptable* they found cigarettes and e-cigarettes on a seven-point Likert type scale ranging from 1 'strongly disagree' to 7 'strongly agree.' They were given two separate statements: 'When thinking about tobacco/e- cigarettes please indicate whether you think they are…….' The order in which e-cigarettes and tobacco cigarette statements were presented was counterbalanced to reduce question order bias.

### Smoking status

Participants were asked to indicate how often (never, occasionally, very often, always) they smoked, vaped or used nicotine products both in the present and the past. Participants were placed in smoking groups based on their 'present' smoking status. Those that answered 'never' to using tobacco or e-cigarettes were defined as non-smokers; those that answered 'occasionally', 'very often' or 'always' were defined as smokers, e-cigarette users or dual users as appropriate.

## Procedure

The study was given ethical approval. The data reported here were used to identify which advertisement would be used in a larger study. The current study and the larger study were funded by Cancer Research UK (CRUK grant number C54622/A20485).

Participants were asked to rate how healthy, desirable and socially acceptable they found tobacco and e-cigarettes before and after viewing 1 of 15 e-cigarette advertisements (see below). In addition, after completing the post-test attitude questions, they were asked to rate the emotional attributes and the perceived effectiveness of the advertisement and to give demographic characteristics. Finally, they were given a debrief about the study, a warning about the addictive nature of nicotine and given links to government quit smoking websites.

## Advertisement selection and preparation

Each participant viewed 1 of 15 advertisements which were allocated randomly in Qualtrics. The 15 advertisements were chosen from a pool of 200 different advertisements displayed online between 2013 and 2016. Ten different themes of advertising were identified which depicted e-cigarettes as being a smoking cessation tool, healthier than (tobacco) cigarettes, aesthetically pleasing, celebrity endorsed, sporty, an alternative to cigarettes in places where cigarettes were restricted, as satisfying, cheaper, more fragrant and as cool as cigarettes.[6] Five researchers coded each advertisement as 1 of the 10 themes and advertisements were chosen that were consistently coded as the same theme by three of the five coders and which the research team found to be the most engaging. The final 15 advertisements included all 10 themes, a variety of brands and images[31] and eight included a smoking or vaping cue and seven did not.[20]

## RESULTS

Observation of table 1 shows that only 2.9% of non-smokers had been full-time smokers in the past and none had been full-time vapers. Only 10% of current e-cigarette users had not smoked in the past; most were either full-time or intermittent smokers previously. There were no smokers who had transferred from daily e-cigarette use to smoking, although 27.8% had used e-cigarettes intermittently in the past. Most dual users had smoked or used e-cigarettes intermittently in the past.

Health scores in the smokers group were not normally distributed due to the positive skew (baseline skewness=2.63, test skewness=2.40). Thus, Wilcoxon signed rank tests were used to compare pre and post intervention scores in each smoking group (smokers, non-smokers, dual users and e-cigarette users) for each outcome measure (health, desirability and social acceptability) of cigarettes and e-cigarettes (see table 2).

## Attitudes towards cigarettes

All smoking groups gave low scores for health of cigarettes at baseline. Showing an e-cigarette advertisement made no significant difference to the score in the smoking, non-smoking and e-cigarette user groups but dual users scored cigarettes as healthier after viewing the advertisement (Z=2.57 p=0.01). There was no change in dual users rating of desirability or social acceptability after viewing an

**Table 1** Percentage of previous tobacco and e-cigarette use of current non-smokers, e-cigarette users, smokers and dual-users

| Previous use | Status | Current non-smokers | Current e-cigarette user | Current smoker | Current dual user |
|---|---|---|---|---|---|
| Tobacco use | Never | 68.5% | 10.0% | 0.9% | 0.7% |
| | Intermittent | 28.5% | 49% | 67.8% | 72.9% |
| | Always | 2.9% | 41% | 31.3% | 26.4% |
| E-cigarette use | Never | 93.9% | 21% | 72.2% | 5.7% |
| | Intermittent | 6.1% | 50% | 27.8% | 88.7% |
| | Always | 0.0% | 29% | 0.0% | 3.6% |

Note: The categories 'occasionally' and 'very often' were collapsed and named intermittent.

advertisement. E-cigarette users scored cigarettes as significantly less desirable (Z=−2.50, p=0.013) and less socially acceptable (Z=−2.501 p=0.012) after viewing an advertisement. Smokers scored cigarettes as less desirable (Z=−2.81, p=0.005) after viewing an advertisement but there was no difference in pre and post scores of social acceptability. Non-smokers showed no change in desirability scores of cigarettes but scored them as less socially acceptable (Z=−4.67 p>0.001) after viewing an advertisement.

Non-smokers scored e-cigarettes as being healthier after viewing an e-cigarette advertisement (Z=2.97, p=0.003), more desirable (Z=2.60, p=0.009) but less socially acceptable (Z=−2.12 p=0.034). E-cigarette users showed no change in attitudes. Smokers scored e-cigarettes as healthier after viewing the advertisements (Z=2.21, p=0.027) but there was no change in desirability or social acceptability. Dual users scored e-cigarettes as being healthier after viewing advertisement (Z=2.53, p=0.011) and more desirable (Z=2.04, p=0.042) but there was no change in social acceptability.

## DISCUSSION

The current study aimed to explore the potential of e-cigarette advertising to (1) enhance attitudes towards cigarettes by reducing negative evaluations towards them

and (2) reduce barriers to the uptake of e-cigarettes by smokers. In terms of renormalisation by changes in attitude, little evidence in the current study supports the notion that viewing e-cigarette advertisements increased positive evaluations of cigarettes. Rather, perceptions of cigarettes being socially acceptable and desirable generally decreased. All smoking groups, other than dual users, showed no change in health scores for cigarettes. However, these groups gave scores close to the scale's floor for health at baseline and there may have been no possibility to reduce scores further after viewing the advertisement. Thus, there was no indication that viewing an e-cigarette advertisement renormalised or encouraged smoking among these smokers, non-smokers and e-cigarette users.

After viewing an advertisement, smokers scored cigarettes as less desirable and e-cigarettes as healthier, supporting the notion that advertisements may reduce barriers to the uptake of e-cigarettes by smokers. Indeed, all groups scored e-cigarettes as healthier after viewing the advertisement, other than e-cigarette users, whose baseline scores were almost at ceiling already. These results show that e-cigarette advertising successfully increases positive attitudes towards e-cigarettes. As perceptions of

**Table 2** Mean healthy, desirability and social acceptability ratings (SD in parentheses) towards tobacco and e-cigarettes by smoking group pre and post advertisement presentation

**Sample**

| Attitude target | Attitude dimension | Non-smoker (n=410) | | E-cigarette user (n=100) | | Smoker (n=115) | | Dual user (n=145) | |
|---|---|---|---|---|---|---|---|---|---|
| | | Pre | Post | Pre | Post | Pre | Post | Pre | Post |
| Cigarette | Healthy | 1.27 (0.83) | 1.28 (0.76) | 1.47 (0.90) | 1.47 (0.81) | 1.59 (0.90) | 1.73 (1.03) | 2.08 (1.35) | **2.26** (1.50) |
| | Desirability | 1.86 (1.45) | 1.77 (1.35) | 2.84 (1.79) | **2.58* (1.67)** | 3.52 (1.78) | **3.21** (1.67)** | 3.85 (1.87) | 3.71 (1.82) |
| | Social Acceptability | 2.63 (1.57) | **2.39*** (1.54)** | 3.04 (1.76) | **2.86* (1.78)** | 3.43 (1.55) | 3.30 (1.64) | 3.36 (1.67) | 3.39 (1.77) |
| E-cigarette | Healthy | 2.53 (1.38) | **2.68** (1.51)** | 4.79 (1.60) | 4.82 (1.65) | 3.03 (1.48) | **3.21*** (1.54)** | 3.80 (1.33) | **4.01** (1.42) |
| | Desirability | 2.76 (1.57) | **2.91* (1.70)** | 5.36 (1.44) | 5.21 (1.37) | 3.91 (1.45) | 3.91 (1.54) | 4.82 (1.21) | **5.03** (1.24) |
| | Social Acceptability | 3.93 (1.58) | **3.81* (1.07)** | 5.08 (1.45) | 5.07 (1.46) | 4.59 (1.39) | 4.50 (1.40) | 5.28 (1.05) | 5.14 (1.18) |

Note: Comparisons are made pre-post for each dimension for each group, with significant differences marked with asterisk(s): *P<0.05; **p<0.005; ***p<0.001.

health are a key determinant in behaviour change, these results encourage the view that smokers may potentially use e-cigarettes as a tool for smoking cessation.[38]

In non-smokers, scores for social acceptability of both cigarettes and e-cigarettes decreased. This result is inconsistent with previous findings which found that when young people viewed e-cigarette advertisements, the ratings for social acceptability increased.[7 29] This discrepancy in results may be due to the different age groups being tested and the suggestion that e-cigarette adverts (and products) are often designed with a younger target audience in mind.[39] It also mirrors a more general negative attitude towards e-cigarettes observed in the UK population in recent years.[27] However, in the current study, after viewing the advertisement, the perception that e-cigarettes are healthy and desirable increased among non-smokers. Although widespread use of e-cigarettes by non-smokers has not been observed to date (in populations where data is available), this result suggests the effects of e-cigarettes advertising on actual e-cigarettes uptake among nicotine naïve users should continue to be monitored, especially in territories where advertising is prevalent.

E-cigarette users showed no change in attitudes towards e-cigarettes after viewing an advertisement. Perceptions of desirability and social acceptability of cigarettes decreased after watching the advertisement. A large proportion of e-cigarette users had previously been smokers so these findings support the view that an e-cigarette advertisement may help deter ex-smokers, who are vaping as a smoking cessation tool, from relapsing.

One group who are relatively under-researched (dual users) displayed a different pattern of responses to cigarettes to the other smoking categories. They rated cigarettes and e-cigarettes as healthier after viewing the advertisement. The contradictory effect of the advertisements on the attitudes of this particular group suggests that the drive to use both tobacco and e-cigarettes is complex. Indeed, some research suggests that many dual users may still use cigarettes because they find it to be a pleasurable experience and use e-cigarettes as a practical solution when smoking cigarettes is banned.[36] Although our initial explorations of this group warrant caution in their interpretation, they do flag the need for further research exploring how dual users perceive and value the differences between e-cigarettes and smoking, and to identify the best strategy to help motivate this group to achieve cessation from combustible products.

### Study limitations
This was an exploratory study and we acknowledge that there were some limitations but that the current work can still contribute to advances in knowledge in the field. As the general public are likely to view a wide range of e-cigarette advertisement types, themes and categories online, we purposely chose to measure changes in attitudes to a variety of advertisements across broad smoking categories rather than assessing the effects of different types of e-cigarette advertisement, for example, cue versus no cue, on specific smoking status. Future research could assess the effects of different types of e-cigarette advertisement on specific smoking status, for example, daily versus intermittent smokers and vapers.

Due to the non-parametric nature of the data, multiple analyses were carried out and thus there was an increased risk of type 1 errors. After deliberation, we chose not to use a statistical method to correct the experiment-wise error rate as this correction may then have increased the risk for type 2 errors. As the purpose of this study was to explore the data with a view to providing a baseline from which to replicate the findings and inform further studies, we considered that making a correction may hinder the accumulation of knowledge in this area and that increased type 1 errors were preferential to increased type 2 errors.[40 41]

In this study, we measured attitudes towards cigarettes and e-cigarettes but did not assess any effects on intention to smoke or actual smoking behaviour. Recent evidence shows that viewing an e-cigarette product[42] or advertisement[43] may act as a cue that leads to an increased urge and desire to smoke or vape. Therefore, future research needs to investigate the relationship between attitude and cue reactivity. The current study had a cross-sectional design and the change in attitude was measured immediately after one exposure to the advertisement so it cannot be determined whether effects of viewing an advertisement would persist over a longer time period. Future research would benefit from assessing changes in attitude longitudinally as well as measuring intentions to smoke or use e-cigarettes and actual smoking behaviour for a fixed period before and after viewing the advertisement.

Our outcomes used single measure attitude scales and although these had high external validity (and single item scales can be regarded as appropriate when constructs are well defined),[44] multiple items scales would have allowed empirical assessment of validity. Furthermore, measuring attitudes towards health using a seven-point Likert type scale resulted in possible floor effects. A more sensitive measure such as a Visual Analogue Scale may have given a broader range of data ensuring that the data were more normally distributed and giving a better indication of changes that may have occurred at the lower end of the scale.

There is a social stigma associated with smoking and so there may have been an effect of social desirability bias. Attitudes in the current study were assessed using explicit, self-report measures so an implicit measure of attitude could be measured in future research as previous research shows that these measures of attitude interact to predict actual smoking behaviour.[45] Additionally, it was considered whether results may be a consequence of the boomerang effect. The advertisements trying to persuade e-cigarettes to be viewed more positively may have inadvertently caused the consumer to resist the persuasion attempt and instead view e-cigarettes more negatively or cigarettes more positively, as a form of non-compliance.

Results were not unexpected, other than for dual users, and so the use of a control group in future research may help to elucidate more of an understanding of attitude changes in response to advertisements in this particular group.

## CONCLUSION

The results from this study suggest that viewing an e-cigarette advertisement is unlikely to renormalise smoking among most groups. However, we observed some evidence that e-cigarette advertisements may increase how healthy cigarettes are perceived among dual users. This highlights the importance of more research with this relatively understudied group, in particular, around factors which promote the decision to transition from dual use to e-cigarette use or nicotine abstinence. In terms of reducing barriers to uptake, e-cigarette advertisement may encourage smokers to quit using e-cigarettes as a smoking cessation tool. Thus, e-cigarette advertisements could be considered a viable tool to stimulate or support smoking cessation. However, where e-cigarette advertising is prevalent, careful monitoring of uptake of e-cigarettes among nicotine naïve populations is important.

**Contributors** DF and IPA conceptualised and designed the study and critically reviewed and revised the manuscript; SC critically reviewed and revised the manuscript; PB drafted the initial manuscript and administered the testing, data collection and statistical analysis. All authors approved the final manuscript as submitted.

**Funding** This work was supported by Cancer Research UK grant number C54622/A20485.

**Competing interests** SC has provided consultancy work for the Pacific Life insurance group on smoking and reduced risk product use and prevalence rates. DF and IPA are both investigators on a randomised controlled trial funded by Allen Carrs Easyway. This trial is comparing the Allen Carr Easway stop-smoking method to local NHS 1-1 stop smoking counselling service (ISRCTN23584477).

**Patient consent for publication** Not required.

**Provenance and peer review** Not commissioned; externally peer reviewed.

**Data sharing statement** Extra data are available by emailing p.booth@uel.ac.uk or fringsd@lsbu.ac.uk.

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
