## [Reviewer comments · BMJ Open]

ARTICLE DETAILS

TITLE (PROVISIONAL)	A survey of the effect of viewing an on-line e-cigarette advertisement on attitudes towards cigarette and e-cigarette use in adults located in the UK and USA: a cross-sectional study
AUTHORS	Booth, Paula; Albery, Ian; Cox, Sharon; Frings, Daniel

VERSION 1 - REVIEW

REVIEWER	Yvonne Chen University of Kansas, USA
REVIEW RETURNED	15-Jan-2019

GENERAL COMMENTS	The authors investigated how an online e-cigarette advertisement changed participants' attitudes toward tobacco smoking and e-cigarette use. The use of samples from two countries is a plus as it gives the field a more nuanced understanding of e-cigarette perceptions in two countries. Below is a list of comments in hopes of helping the authors clarify some minor concerns. Method: --Categorization of participants' smoking status can be more robust. For example, National Youth Tobacco Survey in the U.S. (by CDC) provides more accurate measures to assessing one's smoking status. (There may be similar measures in the U.K. as well.) While this study would not be able to retroactively inquire participants' smoking status, noting that in the limitation section may be helpful.--Could the authors describe the content of the online e-cigarette stimulus? What was featured in the ad, for example? Results: --While the authors found no evidence that normalization (an enhanced, favorable attitude toward cigarettes after viewing an online e-cig ad) occurred, could the decrease in "perceptions of tobacco cigarettes being socially acceptable and desirable" be resulted from the advertising content? For example, if the e-cig ad attached traditional tobacco products, it is not surprising then to find a decrease in positive perceptions among participants. Clarifying the content of the e-cig stimulus can be helpful.--p.13, line 27: The authors stated that "Of all the smoking groups e-cigarette users scored e-cigarettes as the healthiest and desirable before viewing the advertisement and showed no change in attitude after viewing. Perceptions of desirability and social acceptability of tobacco cigarettes decreased after watching the
--

	advertisement. Thus, viewing an e-cigarette advertisement may help deter ex-smokers, who are vaping as a smoking cessation tool, from relapsing.” It seems though that the conclusion is a bit of a stretch based on e-cigarette users’ finding and may be inconsistent with recent cue reactivity studies on how vaping products/ads act as a cue that may entice smokers (See Chen et al. 2018 in Addiction Biology and other recent works). Could the authors clarify this conclusion further while contrasting the findings with recent cue reactivity research? Limitation: --The limitation section is extensive. Given that e-cigarette advertisements have employed a wide variety of appeals (sex appeals, social, quitting, natural, etc.), exposing to only one advertising stimulus may not be comprehensive enough to cover all types of advertising appeals used by the industry. This section could benefit from adding the limitation of having a one-shot experimental design (especially with only one stimulus) to the section. A minor comment: The authors designed an attention check question in its online experiment. While this practice was common among practitioners (and the reviewer certainly has used this method before), a recent Qualtrics study (2017) found that attention check questions in surveys may actually harm survey results. Part of the logic is that participants may fail to pay attention after answering the attention check question. The reviewer would like to share this with the researchers to broaden up the conversations and update the practice: https://www.qualtrics.com/blog/using-attention-checks-in-your-surveys-may-harm-data-quality/
--	--

REVIEWER	Erin Maloney University of Pennsylvania, United States
REVIEW RETURNED	22-Feb-2019

GENERAL COMMENTS	-This is an interesting study aimed at addressing questions that are far from resolved in the public health community. No single study will provide a definitive answer about the effects of exposure to e-cigarette advertising on people’s attitudes and perceptions about e-cigarettes as well as combustible cigarettes. The answer will probably come from compiling the results of a number of these studies, each with their own limitations. That said, I think there is a need to provide much more methodological information about this study in order to make the limitations and appropriateness of the conclusions drawn from the study clearer. -First, the bulk of the conclusions were based on comparisons of participants’ pre-test and post-test scores on ratings of healthiness, desirability, and social acceptability of e-cigarettes within different sub-groups divided out by smoking status. Participants were only exposed to a single e-cigarette advertisement between pre- and post-test assessments. Were the measures used to assess pre- and post-test exactly the same? Were there any foil questions added in? If not, I think this would have some pretty big implications for how people answered the
--

post-test measure in terms of social desirability responses as well as potential boomerang effects. I expected the authors to acknowledge this in the discussion section, but I did not see any discussion about it.

-There is a lot of variability in e-cigarette advertising on a number of variables (e.g., presenting it as an alternative to smoking, a replacement for smoking, or a whole new product unrelated to smoking; inclusion of smoking cues; featuring first generation vs. other generation products). I'm not sure we can generalize about the effects of exposure to e-cigarette advertising in general based on the immediate effects of exposure to one single ad. You can definitely conclude that reactions are different depending upon smoking status, but not necessarily that all ads will produce the same effects. Some information about the content of the advertisement that was used in this study would be extremely helpful. It could provide some insight into why different types of smokers might have reacted differently to the message. Also, considering the sample was drawn partially from the US and partially from the UK, it would be helpful to know if this was an ad that commonly aired in both countries and the brand was popular in both countries. Reactions may be different depending on whether or not the brand and the ad were familiar.

-Were these all single-item measures? Why not use a more established attitude scale or at least use multiple items so that you can provide some information about validity. This should be noted as a limitation to the study.

-Combining participants who responded "occasionally," "very often," and "always" to the question about smoking and vaping habits may be problematic. Previous work found differences between daily and intermittent smokers on similar outcome measures (e.g., Maloney & Cappella, 2015, cited in this paper).

-On p. 11 in the results section the authors noted that dual-users had higher scores on all outcomes at baseline than all other groups. Was a between-subjects test for significant differences conducted? No information is provided.

-Results are confusing to read mostly because A LOT of tests were run. Was there any adjustment for multiple tests? I think results and implications of this study could be clearer if you added a section in the discussion that sought to interpret the findings a bit. Why do you think certain elements changed from pre- to post-test for some groups and not others? Again, this is another place where information about the specifics of the ad would be helpful.

-On p. 13 in the discussion section, the authors state: "Of all the smoking groups e-cigarette users scored e-cigarettes as the healthiest and desirable before viewing the advertisement and showed no change in attitude after viewing. Perceptions of desirability and social acceptability of tobacco cigarettes decreased after watching the advertisement. Thus, viewing an e-cigarette advertisement may help deter ex-smokers, who are vaping as a smoking cessation tool, from relapsing." Do the authors know whether the e-cigarette users in the study were all previously smokers? There is some evidence that those who vape are more likely to take up smoking, so even if participants are not former smokers the finding says good things about exposure to the e-cigarette advertisement used in this study, but it would be

	important to make clear whether or not this group previously smoked. -There is a Booth citation on p. 9 that is not properly formatted. This citation is also not included on the reference page. -There is a Frings citation at the bottom of p. 13 that is not properly formatted. This citation is also not included on the reference page. -Throughout the manuscript, the word “e-cigarette” is used when the sentence calls for the plural form “e-cigarettes.” -P. 7: “Participants were recruited through an online crowd-sourcing tool (Crowdfunder, similar to -MTurk) on which a link to a survey (delivered using Qualtrics) was placed or Twitter.” Should be placed “on” Twitter.
--	---

VERSION 1 – AUTHOR RESPONSE

	Reviewer 1	
	Method	
14	Categorization of participants’ smoking status can be more robust. For example, National Youth Tobacco Survey in the U.S. (by CDC) provides more accurate measures to assessing one’s smoking status. (There may be similar measures in the U.K. as well.) While this study would not be able to retroactively inquire participants’ smoking status, noting that in the limitation section may be helpful.	Thank you for your advice. We recognise there are nuances in smoking status that are potentially missed in our (by necessity) brief measures in the revised discussion, where we suggest that ‘Future research could assess the effects of different types of e-cigarette advertisement on specific smoking status, for example daily versus intermittent smokers and vapers.’ (pg14) In future studies we will give further consideration to different methods of recording smoking status. However, in this study we only wanted to use broad categories to define participants. We have added further detail into the discussion section on page 14. “...we purposely chose to measure changes in attitudes to a variety of advertisements across broad smoking categories”
9	Could the authors describe the content of the online e-cigarette stimulus? What was featured in the ad, for example?	Thank you for your suggestion. We apologise for unintentionally misleading the reader to think that the same image was seen by the participants. In total 15 images were used in the study and each participant saw one randomly allocated image. We have clarified the procedure and expanded the description of the advertisements seen by the participants. See page 9 “Each participant viewed one of fifteen advertisements which were allocated randomly in Qualtrics. The fifteen advertisements were chosen from

		a pool of 200 different advertisements displayed online between 2013 and 2016. Ten different themes of advertising were identified which depicted e-cigarettes as being; a smoking cessation tool, healthier than tobacco cigarettes, aesthetically pleasing, celebrity endorsed, sporty, an alternative to tobacco cigarettes in places where tobacco cigarettes were restricted, as satisfying, cheaper, more fragrant and as cool as tobacco cigarettes.[6] Five researchers coded each advertisement as one of the ten themes and advertisements were chosen that were consistently coded as the same theme by three of the five coders and which the research team found to be the most engaging. The final 15 advertisements included all ten themes, a variety of brands and images[31] and eight included a smoking or vaping cue and seven did not.[20]"
	Results	
9	--While the authors found no evidence that normalization (an enhanced, favorable attitude toward cigarettes after viewing an online e-cig ad) occurred, could the decrease in "perceptions of tobacco cigarettes being socially acceptable and desirable" be resulted from the advertising content? For example, if the e-cig ad attached traditional tobacco products, it is not surprising then to find a decrease in positive perceptions among participants. Clarifying the content of the e-cig stimulus can be helpful.	Thank you for your suggestion. We apologise for giving the impression that only one type of advertisement was used in the study. We have expanded on the description of the advertisements to make it clear that participants saw one of fifteen different advertisements with different appeals (see page 9 for more details) rather than one standard advertisement. Thus, the effect of the advertisement is not linked to a specific type of advertisement content.
15	--p.13, line 27: The authors stated that "Of all the smoking groups e-cigarette users scored e-cigarettes as the healthiest and desirable before viewing the advertisement and showed no change in attitude after viewing. Perceptions of desirability and social acceptability of tobacco cigarettes decreased after watching the advertisement. Thus, viewing an e-cigarette advertisement may help deter ex-smokers, who are vaping as a smoking cessation tool, from relapsing." It seems though that the conclusion is a bit of a stretch based on e-cigarette users' finding and may be inconsistent with recent cue reactivity studies on how vaping products/ads act as a cue that may entice	Thank you for your observation. We acknowledge that with the current wording it seems a bit of a stretch that our findings on the effects of an advertisement to e-cigarette users may infer that it might help ex-smokers from relapsing. We have changed our wording in the discussion and added a Table (see Table 1 page 10) and some text in the results section (see page 10) to show that the majority of e-cigarette users were ex-smokers which may add more support to our inference. "Only 10% of current e-cigarette users had not smoked in the past; most were either full-time or intermittent smokers previously".

	smokers (See Chen et al. 2018 in Addiction Biology and other recent works). Could the authors clarify this conclusion further while contrasting the findings with recent cue reactivity research?	“Perceptions of desirability and social acceptability of cigarettes decreased after watching the advertisement. A large proportion of e-cigarette users had previously been smokers so these findings support the view that viewing an e-cigarette advertisement may help deter ex-smokers, who are vaping as a smoking cessation tool, from relapsing..”pg13 Thank you for drawing our attention to the effects of vaping products/adverts as cues to smoking. We acknowledge that we are unable to discuss whether viewing an e-cigarette advertisement acted as a cue and increased the urge and intention to smoke as we did not assess these variables. We have made this limitation more explicit and cited Chen “In this study we measured attitudes towards cigarettes and e-cigarettes but did not assess any effects on intention to smoke or actual smoking behaviour. Recent evidence shows that viewing an e-cigarette product[42] or advertisement[43] may act as a cue that leads to an increased urge and desire to smoke or vape. Therefore, future research needs to investigate the relationship between attitude and cue reactivity,” pg15
	Limitation	
9	--The limitation section is extensive. Given that e-cigarette advertisements have employed a wide variety of appeals (sex appeals, social, quitting, natural, etc.), exposing to only one advertising stimulus may not be comprehensive enough to cover all types of advertising appeals used by the industry. This section could benefit from adding the limitation of having a one-shot experimental design (especially with only one stimulus) to the section.	We apologise for giving the impression that only one type of advertisement was used in the study. We have expanded on the description of the advertisements to make it clear that participants saw one of fifteen different advertisements (see page 9 for more details) with different appeals rather than one standard advertisement. We have also acknowledged that only viewing an advert on one occasion was a limitation. “The current study had a cross-sectional design and the change in attitude was measured immediately after one exposure to the advertisement so it cannot be determined whether effects of viewing an advertisement would persist over a longer time period.” Page 14.

	A minor comment: The authors designed an attention check question in its online experiment. While this practice was common among practitioners (and the reviewer certainly has used this method before), a recent Qualtrics study (2017) found that attention check questions in surveys may actually harm survey results. Part of the logic is that participants may fail to pay attention after answering the attention check question. The reviewer would like to share this with the researchers to broaden up the conversations and update the practice: https://www.qualtrics.com/blog/using-attention-checks-in-your-surveys-may-harm-data-quality/	Thank you for sharing the blog on attention checks. This was really interesting reading and we will certainly consider the implications of attention checks very carefully when designing surveys in the future.
	Reviewer 2	
9/15/16	-First, the bulk of the conclusions were based on comparisons of participants' pre-test and post-test scores on ratings of healthiness, desirability, and social acceptability of e-cigarettes within different sub-groups divided out by smoking status. Participants were only exposed to a single e-cigarette advertisement between pre- and post-test assessments. Were the measures used to assess pre- and post-test exactly the same? Were there any foil questions added in? If not, I think this would have some pretty big implications for how people answered the post-test measure in terms of social desirability responses as well as potential boomerang effects. I expected the authors to acknowledge this in the discussion section, but I did not see any discussion about it.	Thank you for your comments. Participants were asked for ratings of healthiness, desirability and social acceptability before and after viewing the advertisement. In addition, after viewing the advertisement they were asked to rate the emotional attributes and the perceived effectiveness of the advertisement and then fill in demographic details. Being asked about the effectiveness of the advertisements may partially have acted as a foil. We have changed the wording on page 9 to reflect the procedure more clearly. "Participants were asked to rate how healthy, desirable and socially acceptable they found tobacco and e-cigarettes before and after viewing one of fifteen e-cigarette advertisements (see below). In addition, after completing the post-test attitude questions they were asked to rate the emotional attributes and the perceived effectiveness of the advertisement and to give demographic characteristics." We appreciate your concerns about social desirability and we have acknowledged that this may have had an effect on results in the limitations section on page 15 "There is a social stigma associated with smoking and so there may have been an effect of social desirability bias. Attitudes in the current study were assessed using explicit, self-report measures so an

		implicit measure of attitude could be measured in future research as previous research shows that these measures of attitude interact to predict actual smoking behaviour.[42].” We thank you for drawing our attention to the boomerang effect, we have now mentioned it in our discussion on page 16. “Additionally, it was considered whether results may be a consequence of the boomerang effect. The advertisements trying to persuade e-cigarettes to be viewed more positively may have inadvertently caused the consumer to resist the persuasion attempt and instead view e-cigarettes more negatively or cigarettes more positively as a form of non-compliance. Results were not unexpected, other than for dual-users, and so the use of a control group in future research may help to elucidate more of an understanding of attitude changes in response to advertisements.”
9	-There is a lot of variability in e-cigarette advertising on a number of variables (e.g., presenting it as an alternative to smoking, a replacement for smoking, or a whole new product unrelated to smoking; inclusion of smoking cues; featuring first generation vs. other generation products). I'm not sure we can generalize about the effects of exposure to e-cigarette advertising in general based on the immediate effects of exposure to one single ad. You can definitely conclude that reactions are different depending upon smoking status, but not necessarily that all ads will produce the same effects. Some information about the content of the advertisement that was used in this study would be extremely helpful. It could provide some insight into why different types of smokers might have reacted differently to the message. Also, considering the sample was drawn partially from the US and partially from the UK, it would be helpful to know if this was an ad that commonly aired in both countries and the brand was popular in both countries. Reactions may be different	Thank you for comments. We have now provided a better description of the content of the e-cigarette advertisements that the participants viewed. See page 9. “Each participant viewed one of fifteen advertisements which were allocated randomly in Qualtrics. The fifteen advertisements were chosen from a pool of 200 different advertisements displayed online between 2013 and 2016. Ten different themes of advertising were identified which depicted e-cigarettes as being; a smoking cessation tool, healthier than tobacco cigarettes, aesthetically pleasing, celebrity endorsed, sporty, an alternative to tobacco cigarettes in places where tobacco cigarettes were restricted, as satisfying, cheaper, more fragrant and as cool as tobacco cigarettes.[6] Five researchers coded each advertisement as one of the ten themes and advertisements were chosen that were consistently coded as the same theme by three of the five coders and which the research team found to be the most engaging. The final 15 advertisements included all ten themes, a variety of brands and

	depending on whether or not the brand and the ad were familiar.	images[31] and eight included a smoking or vaping cue and seven did not.[20]" We did some analysis to investigate whether there were statistical differences in changes in attitudes between the different 15 advertisements within each smoking category. However, this entailed a huge number of analyses and there were very few statistical differences and we felt that adding this detail did not really add anything insightful or interesting for the reader.
8/15	-Were these all single-item measures? Why not use a more established attitude scale or at least use multiple items so that you can provide some information about validity. This should be noted as a limitation to the study.	These items were single-item measures. We have clarified this in the measures section on page 8. "Participants rated on three single measures how healthy, how desirable and how socially acceptable they found tobacco and e- cigarettes on a 7-point Likert type scale ranging from 1 'strongly disagree' to 7 'strongly agree.'" Thank you for pointing out this limitation. We have added an acknowledgement of this limitation on page 15 . "Our outcomes used single measure attitude scales and lthough these had high external validity (and single item scales can be regarded as appropriate when constructs are well defined), [44] multiple items scales would have allowed empirical assessment of validity
15	-Combining participants who responded "occasionally," "very often," and "always" to the question about smoking and vaping habits may be problematic. Previous work found differences between daily and intermittent smokers on similar outcome measures (e.g., Maloney & Cappella, 2015, cited in this paper).	Thank you for pointing this out. We have added this issue as a discussion point in the limitations section on page 15 "As the general public are likely to view a wide range of e-cigarette advertisement types, themes and categories online we purposely chose to measure changes in attitudes to a variety of advertisements across broad smoking categories rather than assessing the effects of different types of e-cigarette advertisement, for example cue versus no cue, on specific smoking status. Future research could assess the effects of different types of e-cigarette advertisement on specific smoking status, for example daily versus intermittent smokers and vapers."
11	-On p. 11 in the results section the authors noted that dual-users had higher scores on all outcomes at baseline than all other	Thank you for your question. The statement that dual-users had higher scores on all outcomes at baseline was an

	groups. Was a between-subjects test for significant differences conducted? No information is provided.	observation. In order to make the discussion clearer and to avoid running further multiple analyses we have removed any references of comparisons between groups that were not statistically analysed.
15 14/15/16	-Results are confusing to read mostly because A LOT of tests were run. Was there any adjustment for multiple tests? I think results and implications of this study could be clearer if you added a section in the discussion that sought to interpret the findings a bit. Why do you think certain elements changed from pre- to post-test for some groups and not others? Again, this is another place where information about the specifics of the ad would be helpful.	Thank you for your helpful observations. As you rightfully point out we did use multiple tests and we have added some text to justify this on page 15 "Due to the non-parametric nature of the data multiple analyses were carried out and thus there was an increased risk of Type 1 errors. After deliberation we chose not to use a statistical method to correct the experiment-wise error rate as this correction may then have increased the risk for Type 2 errors. As the purpose of this study was to explore the data with a view to providing a baseline from which to replicate the findings and inform further studies we considered that making a correction may hinder the accumulation of knowledge in this area and that increased Type 1 errors were preferential to increased Type 2 errors. [40,41]" Thank you for this suggestion. We were wary of adding in another section as our discussion is already longer than the suggested number of paragraphs. However, we have changed some of the wording and deleted comparisons that were not statistically analysed to make the results and interpretation clearer. See pages 14/15/16 Thank you for posing this important question. As we showed a wide range of adverts we cannot assign behaviour change to particular advert themes. However, we have changed the wording in places to clarify the differences in behaviour between the smoking groups and given some interpretations.

“After viewing an advertisement smokers scored cigarettes as less desirable and e-cigarettes as healthier supporting the notion that advertisements may reduce barriers to the uptake of e-cigarettes by smokers. Indeed, all groups scored e-cigarettes as healthier after viewing the advertisement, other than e-cigarette users, whose baseline scores were almost at ceiling already. These results show that e-cigarette advertising successfully increases positive attitudes towards e-cigarettes. .” Pg 13

For e-cigarette users...”Perceptions of desirability and social acceptability of tobacco cigarettes decreased after watching the advertisement. A large proportion of e-cigarette users had previously been smokers so these findings support the view that viewing an e-cigarette advertisement may help deter ex-smokers, who are vaping as a smoking cessation tool, from relapsing.”Pg14

One group who are relatively under-researched (dual-users) displayed a different pattern of responses to cigarettes to the other smoking categories. They rated cigarettes and e-cigarettes as healthier after viewing the advertisement. The contradictory effect of the advertisements on the attitudes of this particular group suggests that the drive to use both tobacco and e-cigarettes is complex. Indeed, some research suggests that many dual-users may still use cigarettes because they find it to be a pleasurable experience and use e-cigarettes as a practical solution when smoking cigarettes is banned.[37] Although our initial explorations of this group warrant caution in their interpretation, they do flag the need for further research exploring how dual-users perceive and value the differences between e-cigarettes and smoking, and to identify the best strategy to help motivate this group to achieve cessation from combustible products.. Pg 14

10/14	-On p. 13 in the discussion section, the authors state: “Of all the smoking groups e-cigarette users scored e-cigarettes as the healthiest and desirable before viewing the advertisement and showed no change in attitude after viewing. Perceptions of desirability and social acceptability of tobacco cigarettes decreased after watching the advertisement. Thus, viewing an e-cigarette advertisement may help deter ex-smokers, who are vaping as a smoking cessation tool, from relapsing.” Do the authors know whether the e-cigarette users in the study were all previously smokers? There is some evidence that those who vape are more likely to take up smoking, so even if participants are not former smokers the finding says good things about exposure to the e-cigarette advertisement used in this study, but it would be important to make clear whether or not this group previously smoked.	Thank you for highlighting the need to clarify the previous smoking behaviour of are behaviour so that we can more clearly identify how the advertisements may impact on their current smoking behaviour. We have added a table in the results section on page 10 which shows that a high proportion of the vapers had smoked previously but that no full-time vapers had progressed to smoking. We also added a sentence in the discussion to clarify the previous smoking status of e-cigarette users “A large proportion of e-cigarette users had previously been smokers so these findings support the view that viewing an e-cigarette advertisement may help deter ex-smokers, who are vaping as a smoking cessation tool, from relapsing. Pg 14 We have deleted any statements that had no been statistically analysed “Of all the smoking groups e-cigarette users scored e-cigarettes as the healthiest and desirable before viewing the advertisement.”
	-There is a Booth citation on p. 9 that is not properly formatted. This citation is also not included on the reference page.	Thank you for identifying this citation. This has now been removed as the section has been rewritten.
	-There is a Frings citation at the bottom of p. 13 that is not properly formatted. This citation is also not included on the reference page.	Thank you for identifying this citation. This has now been removed as the section has been rewritten.
	-Throughout the manuscript, the word “e-cigarette” is used when the sentence calls for the plural form “e-cigarettes.”	Thank you for your observation. I have changed e-cigarette to its plural form when appropriate.
	-P. 7: “Participants were recruited through an online crowd-sourcing tool (Crowdfunder, similar to -MTurk) on which a link to a survey (delivered using Qualtrics) was placed or Twitter.” Should be placed “on” Twitter.	Thank you for pointing out this typo. This line has now been edited out of the final version.